# Harnessing Digital Health Technologies to Remotely Manage Diabetic Foot Syndrome: A Narrative Review

**DOI:** 10.3390/medicina57040377

**Published:** 2021-04-14

**Authors:** Bijan Najafi, Ramkinker Mishra

**Affiliations:** Interdisciplinary Consortium on Advanced Motion Performance (iCAMP), Michael E. DeBakey Department of Surgery, Baylor College of Medicine, Houston, TX 77030, USA; ram.mishra@bcm.edu

**Keywords:** foot disease, diabetic foot, ulcer, amputation, remote patient monitoring, wearable, digital health, telehealth, diabetic peripheral neuropathy, care in place

## Abstract

About 422 million people worldwide have diabetes and approximately one-third of them have a major risk factor for diabetic foot ulcers, including poor sensation in their feet from peripheral neuropathy and/or poor perfusion to their feet from peripheral artery disease. The current healthcare ecosystem, which is centered on the treatment of established foot disease, often fails to adequately control key reversible risk factors to prevent diabetic foot ulcers leading to unacceptable high foot disease amputation rate, 40% recurrence of ulcers rate in the first year, and high hospital admissions. Thus, the latest diabetic foot ulcer guidelines emphasize that a paradigm shift in research priority from siloed hospital treatments to innovative integrated community prevention is now critical to address the high diabetic foot ulcer burden. The widespread uptake and acceptance of wearable and digital health technologies provide a means to timely monitor major risk factors associated with diabetic foot ulcer, empower patients in self-care, and effectively deliver the remote monitoring and multi-disciplinary prevention needed for those at-risk people and address the health care access disadvantage that people living in remote areas. This narrative review paper summarizes some of the latest innovations in three specific areas, including technologies supporting triaging high-risk patients, technologies supporting care in place, and technologies empowering self-care. While many of these technologies are still in infancy, we anticipate that in response to the Coronavirus Disease 2019 pandemic and current unmet needs to decentralize care for people with foot disease, we will see a new wave of innovations in the area of digital health, smart wearables, telehealth technologies, and “hospital-at-home” care delivery model. These technologies will be quickly adopted at scale to improve remote management of diabetic foot ulcers, smartly triaging those who need to be seen in outpatient or inpatient clinics, and supporting acute or subacute care at home.

## 1. Introduction

The prevalent and long-neglected diabetic foot ulcer (DFU) and accompanying lower extremity complications rank among the most debilitating and costly sequelae of diabetes in both the developed and developing world [1]. Globally, it is estimated that 18.6 million (15.0–22.9) people currently have an active DFU; an additional 131 million (1.77% of the global population) have precursor risk factors in developing a DFU without intervention [2]. It is estimated that one-third of all diabetes-related costs are spent on diabetic foot care in the United States, with two-thirds of these costs incurred in the inpatient settings, constituting a substantial cost to society [1,3]. In 2012 alone, the total costs of diabetes were USD 245 billion, a 41% increase since 2007 [4]. A previous report by Skrepnek et al. [3] included over 1 million cases of DFUs that presented to emergency departments (EDs) in the US from 2006–2010 and suggested a national cost of USD 1.9 billion per year for ED treatment of DFUs alone, and USD 8.78 billion per year in costs for inpatient care alone. Perhaps no complication of DFU is more significant than lower extremity amputation (LEA), which occurs at a rate of 10% per DFU. Globally it is estimated that 6.8 million people had an amputation because of DFU; among, them 4.3 million (3.7–4.9) do not yet have a prosthesis [2]. Hispanics and African-Americans appear to be at increased risk for LEA compared to other ethnic groups and may also be less likely to undergo advanced care to prevent limb loss. Further, in low-income populations, the risk of major LEA is estimated to be 38% higher compared to the highest income regions (*p* < 0.05) [3]. These data suggest an important gap in effectively managing DFUs, particularly among the working poor, and is most important for Hispanics and African-Americans.

The consequences of DFU are not limited to amputation. DFU may put patients at risk for other adverse events such as falls, fractures, reduced mobility, frailty, and mortality [5,6,7]. For example, mortality after amputation because of diabetes is estimated to be 70% at 5 years, which exceeds many common cancers such as breast cancer and prostate cancer [8]. Even if DFU is successfully treated, patients may often suffer from significant lower extremity muscle atrophy, particularly if irremovable offloading was used for more than 4 weeks [6], leading to premature frailty and reduced mobility.

In light of the impending diabetes epidemic and high prevalence of DFU and its associated complications, the need for enhanced prevention of DFUs is clear. At least 70% of amputations are potentially preventable [9]. Randomized controlled trials (RCT) and meta-analyses show that DFU is preventable by controlling these key reversible risk factors using interventions such as appropriate foot care, footwear, daily monitoring plantar temperature, and medical management [10,11,12,13]. However, effective technology is still missing to facilitate monitoring DFU related risk factors on a daily basis, empower patients in self-care as well as engaging them to use these technologies, and effectively coordinate care among circles of care providers and caregivers. Digital health technologies could fill the gaps and speed up the translation of these effective interventions to become clinical standards for the management of DFUs.

This review paper summarizes some of the promising developments in the area of digital health that may promote the prevention and management of DFU. In particular, we are focusing on the technologies that can facilitate care in place or empower self-care for patients with active or high risk of DFU. More specifically, recent developments in three specific areas are discussed (shown in detail in Figure 1): (1) Triaging high-risk patients for timely intervention, (2) Care in place to avoid frequent hospital visits, and (3) Self-care to empower patients and their caregivers.

## 2. Triaging High-Risk Patients

Typically, a diabetic person at high risk for DFU visits an outpatient clinic weekly or bi-weekly for regular podiatry evaluation [10]. However, frequent visits could easily overwhelm the existing healthcare system, which is already overburdened. Even specialty diabetes centers with dedicated staff and top-shelf resources are often not enough, and the ulcer recurrence rate is still alarmingly high [14]. Therefore, identifying and referring high-risk patients is of utmost priority for providing timely intervention to those at the edge of foot ulceration.

### 2.1. Early Detection of DFU

The recurrent trauma at pressure points on the sole increases the risk of local inflammation, which an elevated temperature can detect at the affected site [15,16]. Prior studies suggest that a persistent temperature difference between identical sites on opposite feet, exceeding 2.22 °C (4.0°F), on two consecutive days can accurately predict ulcer development [16,17,18]. Therefore, a diabetic foot prevention program incorporating foot temperature monitoring may lower foot ulcer recurrence rates in high-risk patients [19].

Despite inclusion in several clinical practice guidelines [20,21,22], only four studies investigated the efficacy of routine home monitoring of plantar temperatures to prevent DFUs [23,24,25,26]. Among these, three studies were conducted by the same research group with the recruitment of 427 participants. In these studies, participants assessed temperature at 12 locations on their feet using an infrared thermometer [23,24,25,26]. Overall, results demonstrated a significant reduction in foot ulcer incidence in those allocated to remote foot temperature monitoring than standard care. However, the fourth study with 41 patients, performed by the different groups, reported no significant effect of home foot temperature monitoring [26]. Post hoc power analysis showed that inconsistent results could be due to the high risk of bias attributed to low sample size and imbalance between the group for the number of patients with multiple ulcer histories [11].

Furthermore, mostly a handheld device, such as an infrared thermometer, is used to assess the foot temperature, which can be cumbersome and time-consuming for patients with diabetes. To improve self-care and patient active participation, there is a need to automate home foot temperature monitoring. In 2017, Frykberg et al. developed a wireless thermometric foot mat called Podimetrics mat (Figure 2A) to encourage the adoption of daily foot-temperature monitoring in patients with diabetes and prior DFU [16]. Briefly, when a person steps on the mat for about 20 s, it automatically takes a thermogram of both feet. The thermogram accurately measures temperature over the range of 15 to 40 °C and transmits the data securely to an approved server managed by the manufacturer. In this feasibility study, the authors demonstrated a temperature difference of 2.2 °C between common sites on both feet correctly predicted 97% of foot ulcers, with an average lead time of 37 days and a specificity of 57% among 129 patients with diabetes and prior DFU. Later using Podimetrics, Lavery et al. (2019) took an innovative approach for early detection of DFUs by automation of unilateral foot temperature measurement [18]. The benefit of this approach was in being able to detect the risk of DFU in patients with a wound to one foot and those with proximal lower extremity amputation as the conventional method required comparison of temperatures between contra-laterally matched anatomy, limiting its use in high-risk cohorts.

Moreover, Najafi et al. (2017) tested Smart Socks, an optical-fiber-based textile that measures plantar foot temperature, plantar pressure, and toe range of motion (Figure 2B). They found a moderate agreement (r = 0.58) in foot temperature changes between Smart Socks and a thermal infrared camera [27]. The main benefit of this technology lies in its applicability in everyday life and ability to simultaneously assess potential indicators (e.g., plantar pressure and toe range of motion) of DFUs. However, the validity of Smart Socks to predict prospective DFU remains to be studied. While there is an ongoing debate about the advantage of continuous monitoring over single-point measurement (e.g., as assessed by Podimetrics Mat [29]), earlier results from Najafi et al. study suggest that continuous and simultaneous measurement of temperature and pressure may assist in determining less appreciated but important digital biomarkers of DFU, such as thermal stress response, which is associated with shear [30] and vascular health [31] and may be a strong predictor of foot complication like acute Charcot foot [32].

### 2.2. Remote Monitoring of DFU Risk

Recently, developed innovative technologies can remotely measure early digital biomarkers of DFU, which can provide screening tools with high sensitivity and facilitate in triaging patients at high risk of DFU. These technologies take into account that most foot ulcers occur due to repetitive trauma at the pressure points on the sole for several days [10]. Due to unchecked repetitive trauma, plantar tissue stress (PTS) elevates in the foot area exposed to high pressure [13]. PTS represents various mechanical factors, including plantar pressure, shear stress, and time spent without protected footwear (adherence). A persistently elevated PTS level can result in sub-dermal inflammation and eventually a DFU [13]. A prior meta-analysis suggests that people with Diabetic peripheral neuropathy (DPN) and a history of foot ulcers have higher plantar pressures during walking than those with DPN who have not had an ulcer [33]. The assessment of PTS is currently based on pressure plates or insoles with pressure sensors located within health care or research facilities [34]. However, the existing technologies provide a snapshot of the PTS, which is insufficient to timely alert when the plantar pressure is too high to prevent the incident of DFU.

To address this gap, recently, smart insoles have been designed with sensors that remotely monitor continuous plantar pressure and provide the user with alert-based feedback when plantar pressures are too high [28,35,36]. In a typical smart insole system, there are two primary components: (1) a pressure sensor-embedded insole, which converts plantar pressure into an electrical signal, and (2) a transmitter that triggers an alert in the form of audio, visual, or tactile feedback [37]. Pressure sensors are generally placed on three positions under the metatarsal heads, two under the lateral plantar surface, one under the heel, one under the hallux, and one under the lesser toes.

Other digital health products available in the market exist to facilitate home monitoring of plantar pressure and gait. For example, Surresense Rx (Orpyx, Calgary, Canada) uses a smartwatch to monitor and notifies sustained plantar pressure (pressure above 30–50 mmHg lasting for longer than 15 min) during activities of daily living. In 2019, Abbott et al. [38] tested the efficacy of this technology in preventing the recurrence of DFU (Figure 2C). In summary, they recruited 90 patients with a history of DFU and randomized them to the intervention group (IG) and control group (CG). The IG received a functional smartwatch that alerts when a sustained pressure (pressure lasting longer than 15 min) is detected, while no feedback was provided to the CG. They followed all participants up to 18 months. Results suggested a 71% reduction in ulcer recurrence thanks to timely and interactive feedback provided to the IG. However, the high dropout rate (approximately 35%), small sample size, and the small number of DFU cases limit the rigor and, consequently, the conclusion of this study. Despite these limitations, this study probably could be a significant step forward supporting the use of digital health technology to engage patients in preventive self-care. The same company designed Orpyx LogR that uses a smartphone to monitor and remotely visualize (via the cloud) in-shoe plantar pressure with an autonomy lasting approximately 8–12 h of active use. However, the clinical validity of this platform is remained to be studied.

### 2.3. Remote Monitoring of Wound Characteristics

In the last decade, mobile phones have transformed from a simple communication device to a smart technology used for various tasks such as GPS navigation, internet browsing, gaming, messaging, or video calls. Thanks to these transformations, there is exponential growth in smartphone-based applications for remote monitoring of wounds and empowering patients in taking care of their chronic conditions. In 2015, Wang et al. designed and implemented a smartphone-based wound image analysis system for patients with DFUs that could detect wound boundaries and determine healing status [39]. Similarly, Mammas et al. conducted a simulation-based experiment and asked ten specialists to examine a DFU remotely to evaluate a mobile-telemedicine platform’s feasibility and reliability [40]. They demonstrated that the platform allowed the wound’s remote classification with an average accuracy of 89% and acceptability of 89–100% among specialists. The use of mobile images for characterizing wounds is, however, debated. For example, van Netten et al. found mobile phone images had low validity and reliability for remote assessment of DFUs and recommended that clinicians use additional information to make treatment decisions when using mobile phone images [41]. While the smartphone-based patient care technologies are in the early stage, future studies are needed to improve image quality, determine wound characteristics, and manage the patients’ electronic health records.

### 2.4. Smart Patient Referral

Many health care quality improvement experts recommend improving the process of high-risk foot care through stratified foot risk exams [42]. These exams have been shown to be useful to prevent LEA of DFU up to 70% [43]. However, currently available technologies remain insufficient to be used on a routine basis by non-expert clinical staff. Unfortunately, the reality is that it is difficult to translate conventional multifactorial risk stratification models from largely private, tertiary care academic centers to community clinics, which provide care to low-income populations. The centralized center “last resort” referral model is also inherently flawed because central multidisciplinary diabetic foot and wound clinics become easily overloaded when already over-taxed staff and resources become inundated with critically ill patients. This overflow produces scheduling backlogs, increases emergency department (ED) visits and hospitalizations, and leads to untimely LEA due to lack of coordinated outpatient risk stratification and prevention of DFU, where intervention should be simpler, effective, and low cost. To fill the gap, new technologies are desperately needed to assist in better risk stratifications without installing costly infrastructure or training dedicated staff. In an effort to improve ease of classification of plantar wounds, determine the risk of lower extremity amputation, and triage those who could benefit from advance care like revascularization, Mills et al. [44] suggested a new wound classification named Wound, Ischemia, and foot Infection (WIfI), which classifies wound based on three major factors that impact amputation risk and clinical management: WIfI. This classification has also been adopted by the Society for Vascular Surgery under the name of “The Society for Vascular Surgery’s Threatened Limb Classification” and integrated with a mobile application, called “SVS iPG” which is freely available for download on Apple App Store or Google Play. The SVS iPG application (Figure 3) provides education on different wound types and includes guidelines to manage each wound type, including the recommendation for referral to vascular surgeons. In addition, it includes a calculator to estimate the WIfI score for DFU based on easy-to-assess metrics associated with wound size, ischemia, and degree of infection (Figure 3). Figure 3 illustrates a typical example of the WIfI score based on entered information and provides an estimation for risk of LEA as well as a recommendation for the benefit of revascularization.

Recognition of infection and ischemia is very important to determine factors that predict the healing progress of DFU and decision making for a referral to specialized multidisciplinary wound centers. Patients with an active DFU and particularly those with ischemia or gangrene should be checked for the presence of infection. Approximately 56% of DFU become infected, and 20% of DFU infections [45,46]. If the infection progresses, many patients require hospitalization and, most likely surgical resections or amputation [46]. For instance, Skrepnek et al. demonstrated that across 5.6 billion ambulatory care visits between 2007 and 2013 in the US, 784.8 million involved diabetes and DFU, and associated infections constitute a powerful risk factor for emergency department visits and hospital admission [47]. Due to high risks of infection and ischemia in DFU leading to patient’s hospital admission and amputation [48], recognition of infection and ischemia in DFU with cost-effective machine learning methods is a very important step towards the development of a complete computerized DFU assessment system for remote monitoring in the future. Recently, few efforts have been made to automate the DFU pathology recognition using foot photographs. To date, the most relevant research in this field is by Goyal et al. [49]. They proved that deep learning methods outperformed conventional machine learning methods on a small dataset (1459 images) and proposed an Ensemble convolutional neural network (CNN) approach for ischemia and infection recognition. Although they achieved high accuracy in ischemia recognition, there are a few limitations: The proposed binary classification Ensemble CNN method detected one class at a time, which cannot detect the co-occurrence of infection and ischemia;The dataset is small and cannot be generalized; andThe recognition rate of infection is 73%, which requires substantial work to improve the accuracy. Despite these efforts on still in infancy, we anticipate that with advances in artificial intelligence (AI) and advanced analytical approaches, a more accurate computerized method would be emerged to smartly automate the DFU pathology recognition using mobile apps and by non-wound care specialists. Such development could facilitate smart triaging of patients with DFU who could benefit from hospital referral for revascularization or advanced wound care management.

## 3. Care in Place

Long before the COVID-19 pandemic disrupted care delivery to patients with acute or chronic illness, new innovative solutions were emerged to provide care for the subset of patients who could receive hospital-level medical services in the comfort of their own homes [50,51]. This pandemic shows that traditional healthcare delivery models for managing a chronic illness like DFU are not at scale to handle situations like the global COVID-19 crisis. Because people with diabetic feet represent a fragile population, they have been urged to avoid unnecessary hospital admissions and outpatient visits to reduce their exposure risk to COVID-19. This has disrupted best practices for preventing disease-related complications [52]. In response, many healthcare providers are re-engineering their pathways to promote “care in place”. Care in place is an increasingly important topic in health care, becoming the foreground in governance practices to decentralize care delivery and reduce care disparities, particularly for people not living in city populations or living in disadvantaged zip code areas [53]. For example, the latest foot disease guidelines emphasize that a paradigm shift in research priority from siloed hospital treatments to innovative integrated community prevention is now critical to address the high foot disease burden [54] through policies of concentration and decentralization, becoming a focal point for policy-makers, managers, professionals, and patients. Thanks to connected technologies, new solutions have emerged that have significant potential to address some of the current healthcare system’s greatest challenges and, importantly, may result in better health outcomes for citizens while reducing the financial burden on the public purse. Figure 4 illustrates several categories of care in place that will be briefly discussed in this section.

### 3.1. Telemedicine Visits

Telemedicine is the most appropriate solution for the remote delivery of care. Telemedicine is defined as “practicing medicine at a distance,” in which healthcare providers examine, observe, and treat a patient from a remote location [29]. The objective of telemedicine is to shift care from hospitals and clinics to homes, which might even be safer for the patients. Recent advancement in telecommunication systems has benefitted telemedicine to emerge as one the most economical and patient-friendly methods for delivering follow-up care to patients with DFUs [55]. Furthermore, the COVID-19 pandemic has promoted patient acceptability and accelerated the adoption of telehealth/telemedicine in hospitals and outpatient clinics [56].

Currently, telemedicine for wound care is mainly focused on assessments of wound images [29]. We summarized telemedicine-related studies related to DFU care in three categories: (1) efficacy, (2) cost, and (3) reliability (Figure 5). In a landmark study, Rasmussen et al. (2015) evaluated telemedicine’s efficacy by comparing the telemedical and standard outpatient monitoring in the care of patients with DFUs, focusing on ulcer healing and amputation [57]. They included 401 patients with DFU and followed them for one year or till healing or amputation or dead, whichever came first. One hundred ninety-three participants in the IG received a cluster of two telemedicine consultations with one visit to the clinic, while the CG with 181 patients received three visits in the clinic. They continue this process until DFU healed or an adverse event occurred. The study found no difference in healing and amputation, which suggests that telemedicine is at least as useful as routine clinic visits. However, there was a higher mortality rate in the group with telemedicine, which is hard to interpret.

Previous studies also evaluated the clinical effects of telemedical monitoring of DFU, but they were either low-powered or included ulcers of mixed etiology and different approaches to telemedicine, making it difficult to compare them [60,61]. In a nonrandomized study, Wilbright et al. concluded that the effectiveness of real-time interactive telemedicine consultation in managing DFU among 20 patients was equivalent to ulcer care at a diabetes foot program with in-person visits for 120 patients [61]. Furthermore, in 1997, reimbursement-related issues forced 2700 home care agencies to close, which overburdened the remaining home care agencies and adversely affected the healing rate due to inadequate care. Kobza et al. (2000) arranged telemedicine-based video visits for the patients and assessed healing rate, number of health visits, and number of hospitalizations related to wound complications. They reported an improved healing rate and reduced hospitalizations with telemedicine compared to baseline home care visits [60].

To reduce leg ulcer-related economic costs, Summerhayes et al. (2012) used a telemedicine system that provided images of the wound to the primary and secondary healthcare professionals, allowing them to monitor wounds and design the treatment plan [62]. The author estimated total health care cost by combining primary (i.e., leg ulcer clinic charges, salaries of the leg ulcer nursing team, and clinic room cleaning costs) and secondary costs (i.e., charge to the primary care trust for secondary care outpatient appointments and the cost of any procedures performed) in a semi-rural medical practice in the UK. Their results showed that telemedicine reduced leg ulcer care’s total cost by 11.9% and promoted faster wound healing from 105 days to 70 days compared to conventional care. Therefore, they encouraged introducing the telemedicine system for leg wound care to local general practices across the region. In another study, Fasterholdt et al. (2018) found that telemonitoring costs were 41% lesser than standard monitoring while the amputation rate was the same; however, this difference was not statistically significant [58].

Moreover, to determine the telemedicine system’s reliability, Bowling et al. (2011) compared clinical assessment of wounds using direct visits done by two clinicians and remote evaluation of wound images collected using a 3D wounds imaging system rated by three clinicians [59]. However, among patients who needed debridement, they found most disagreements and linked it to the imaging system’s inability to pick up a few wound characteristics such as moistness.

To address these gaps, innovative wearable technologies have emerged, such as smart dressing that could provide additional information about wound characteristics. These dressings enable monitoring bacteria in the wound dressing [63]; wound-site temperature, sub-bandage pressure, and moisture level from within the wound dressing [64]; wound pH [65] and even physiological stress as a potential indicator of delay in wound healing [66]. Mehmood et al. proposed a low-power portable telemetric system for wound condition sensing and monitoring, which enables measuring and transmitting real-time information of wound-site temperature, sub-bandage pressure, and moisture level from within the wound dressing [64].

### 3.2. Care Coordination

Diabetes-related comorbidities, such as hypertension, hyperlipidemia, chronic kidney disease, or cardiovascular disease, often complicate diabetes care [67]. While diabetes specialists (e.g., podiatrists, dieticians, ophthalmologists, or endocrinologists) are available to treat each of the diabetes-related comorbidities, lack of coordination among the healthcare professionals can lead to fragmented healthcare. In the overwhelming majority of instances, clinicians are unaware of a patient’s history and ongoing treatment of the comorbidities [68]. Additionally, due to the intricacies of diabetes care, patients may self-report inaccurate information with the increase in comorbidities [69] and often do not understand how to take care of themselves at home [70]. To address these issues, a well-designed care coordination model can provide a viable solution.

Care coordination is referred to deliberately organizing patient care activities and sharing information among the stakeholders (including medical professionals, patients, and caregivers) to facilitate the appropriate delivery of health care services [69]. Failure in care coordination can lead to medication errors, hospital readmissions, avoidable emergency department visits, duplicate testing, and disease progression from inadequate delivery of preventive services [71]. In contrast, well-coordinated care ensures effective communication between provider-to-provider and provider-to-patient for better disease management [72,73]. For instance, Chen and Cheng (2020) demonstrated a lower risk of hospitalization for diabetes-related conditions among patients treated by physicians with a greater number of shared patients as this allowed better-coordinated care [72].

Over the last decade, advancement in Health Information Technology (HIT) has made it possible to efficiently record and share medical information with relevant people through electronic health records (EHR) [74]. An EHR is the health information electronically stored systematically, which can provide required information instantly and securely to the authorized users. Several studies reported adoption of EHR reduced hospitalization, improved clinical biomarkers, and reduced the cost of care for patients with diabetes, which stem from enhanced coordination between health care providers [74,75,76]. In 2009, Health Information Technology for Economic and Clinical Health (HITECH) Act incentivized hospitals with USD 30 billion to adopt EHR systems. Consequently, ninety-six percent of acute care hospitals in the US have adopted EHR technology by 2015, up from just 9% in 2008 [77].

There are two ways in which EHR-enabled innovations can be used to improve diabetes care. First, diabetes registries formed using data extracted from EHRs for patients with diabetes and use a standardized method to organize the information such as patient demographics, laboratory results, pharmacy data, and comorbidity data [78]. A well-designed patient registry allows identifying patients at high risk to deliver targeted treatment and increases provider’s adherence to clinical guidelines [79,80]. In an RCT, patients with outlying clinical biomarkers and overdue follow-up were identified from a diabetes registry with 3079 patients, and reminder letters were sent to the patients for the testing while primary care providers were alerted via email about the patients at risk [79]. While the overall quality of care improved in the IG due to improved patient–provider engagement, there were no significant between-group differences in glycemic control. In another study, patients with poor baseline glycemic control were identified from a diabetes registry. Patients in the IG received an email with a quarterly report card from the healthcare provider. Although patients in the IG and provider well-received the quarterly report card and proportion of patients (6.4% increase vs. 3.8% increase in the CG, *p* < 0.001) increased significantly, there was no improvement in clinical outcomes [80]. Additionally, Joret et al. showed that a multidisciplinary clinic with good coordination between the healthcare providers could reduce diabetic foot-related costs and improve patient outcomes [81].

Furthermore, studies have shown that patient portals linked to a patient’s EHR can improve patient-provider communication [82]. For instance, in an RCT, Ralston et al. provided patients with diabetes access to a web-based portal (linked to EHR) for 12 months through which they exchanged information with the provider, got feedback on blood glucose readings, maintained an online diary for physical activity and nutrition, and got access to an educational site [83]. Results showed an average decline in HbA1c levels by 0.7% (95% CI: 0.2–1.3) among patients in the IG compared with the CG. Patient portals allow patients with diabetes to exchange information (e.g., foot images, daily-glucose level, or online diary for physical activity and nutrition) with their health care provider, request for refills of prescription medications, schedule an appointment, and access diagnostic test, which can improve patient’s self-management and better engage with healthcare provider [84,85,86].

While EHR systems are rapidly integrating with the existing healthcare setup, there are issues associated with EHR, such as lack of interoperability, complicate to use, and costly to configure. In recent years, AI has offered data-driven solutions to answer clinically meaningful questions using EHR and make EHRs more user-friendly [87,88]. To address these issues, various studies have demonstrated AI can be used to identify diabetes phenotype [89], monitor progression of diabetes [90] and hypoglycemia [91], and real-time wound assessment [92]. However, due to a lack of explanation about AI-based system suggestions, physicians are reluctant to rely on AI-based assessment of EHR entirely. To improve compliance of primary care providers (PCP) with the EHR system, new startups like Epic, Cerner, Allscripts, and Athena are bringing AI capabilities such as natural language processing, explainable AI, and automated image analysis [93].

Clinical decision support (CDS) is the second EHR-enabled innovation that can enhance patient–provider interactions to translate the information from diabetes registries into clinical action [78]. CDS systems can provide clinical alerts to the healthcare provider during the patient visit, improving the provider’s adherence to clinical guidelines [94]. These clinical guidelines may involve reminders for foot examination, routine lab testing, recommendations for specific medication choices, and alerts for potential drug–drug interactions [74]. Despite the well-documented benefits of preventive care in the early detection of foot ulcers, in 2014, the Center for disease control and prevention (CDC) reported that one-third of the patients with diabetes did not receive annual foot examination from their PCP. To improve PCP’s adherence to foot examination, William et al. used the CDS system to alert PCP to perform a comprehensive diabetic foot exam in the clinical setting. After implementing the reminder system in the EHR, reporting of diabetic foot examination increased from 4% to 78% [95]. Other studies demonstrated that the CDS system could improve preventive healthcare by prompting the PCP to educate patients about the daily self-foot assessment during the clinic visit, thus improving patient-provider engagement [96,97,98].

### 3.3. Hospital at Home

The hospital at home (HaH) program is getting popular amid the pandemic. HaH is a patient-centric healthcare model in which acute care is provided to the patients outside the hospital setting. Previous studies have shown that the HaH care is feasible [51], acceptable, and effective in providing hospital-level care to patients at home with a 21% reduction in mortality and 24% reduction in readmissions [99]; and has on average 11% lower cost compared to equivalent hospital care [100]. Thanks to recent advances in the field of digital health, telehealth, and the internet of things (IoT), innovative solutions have emerged that can provide medical care to the subset of patients in the comfort of their homes (Figure 4) [13]. An efficient HaH program would facilitate a successful transition from inpatient to outpatient, reduce the risk of readmission due to diabetes-related complications, and bring down the burden from family caregivers and hospitals.

Furthermore, healthcare providers prefer HaH care for patients with diabetes as it allows them to be ambulatory in a safe environment. Over the last decade, the range and level of complexity of medical devices used in the home have increased dramatically. Now medical equipment, such as home dialysis, insulin delivery pump, apnea monitors, pulse oximeter, can be used by the family-care givers at home [101]. Furthermore, care recipients can independently use diagnostic and testing devices to monitor their cholesterol and blood glucose levels at home.

To manage acute diabetes, HaH physicians or experienced practice nurses are available 24 h through the virtual meeting or regular/intermittent home visit as required. These nurses play a vital role in-home care as they can perform relatively complex medical procedures such as provide home infusions, medication reconciliation and management, educating patient and caregiver, wound care, and other nursing procedures. However, the needs of nursing services in the outpatient settings are primarily related to managing visiting hours of healthcare professionals, allocating care providers based on patient’s needs, and training of care-provider. To overcome these challenges, hospitals are using data analytics technologies. In Minnesota, Allina Health, a nonprofit health system, uses data analytics software developed by AMN Healthcare to manage staff hours and costs automatically.

Although there is no evidence-based study assessing HaH care’s benefit over usual care in the context of diabetes, there is a rapid growth in professional home care services. For instance, BestBuy Health, a leading provider of technology products, is expanding its healthcare digital wellness efforts and moving from selling wearables and devices to adding services needed to help patients with chronic illness [52]. Similarly, Medically Home, a Boston-based company, offers an integrated technology platform and network of in-home services under Mayo Clinic physicians and providers’ guidance. We anticipate pandemic will accelerate the adoption of the HaH care delivery model. Moreover, home-based medical care will be mainstreamed into the US healthcare delivery system for wide ranges of acute and subacute conditions, including managing severe cases of DFU [52].

## 4. Technologies to Empower Self-Care

Preventing DFU requires the patient to have a central active role in self-managing their health [10,102]. This is a shift from current practice, in which frequently the patient is a passive recipient of care. Patient behavior change support can help achieve self-management but is not currently available [103,104]. This section summarizes few recent developments which could engage patients in a healthy ecosystem as an active recipient of care.

### 4.1. Wellness Program

#### 4.1.1. Exercise

Physical activities can help in preventing DFUs by improving blood glucose control [105], blood perfusion in upper and lower extremities [106], and overall well-being [107]. Despite the well-known benefits of regular physical activity, only 25% of adults with diabetes comply recommended amount of physical activity, and 40% are physically inactive, engaging in less than ten minutes of moderate or vigorous activity per week during work, leisure time, or transportation [108]. Exergames, gamification of physical activity using technology, offers an enjoyable and promising new approach that can increase physical activity and promote exercise adherence in individuals with diabetes [109]. For instance, Höschman et al. used a novel smartphone game to deliver individualized exercise and physical activity promotion among 18 people with DM2 (mean age 57) for 24 weeks [110]. The game was based on a storyline to be more engaging with the theme presenting the restoration of a garden as a metaphor for one’s own body and the taming of the Schweinehund, which in German represent a self-deprecating idiom denoting one’s weaker self, often referring to the lazy procrastination regarding PA [110]. Despite an average increase of 3998 steps/day, there was no change in glycemic control (HbA1c) throughout the intervention [110]. However, an earlier study that used a Wii Fit Plus sports game for 12 weeks 30 min a day improved adherence to physical activity and decreased HbA1c from 7.1% to 6.8% among 93 participants in the IG [111]. Therefore, the results of Höschman et al. (2019) could be attributed to insufficient exercise intensity, which might have failed to elicit an improvement in HbA1c.

Furthermore, often, people with DFUs are in a poor physical condition and have medical complexity due to which weight-bearing exergames may not provide a feasible solution. For example, the uptake of exercise is limited in patients with end-stage renal disease undergoing hemodialysis. To address this gap, Zhou et al. demonstrated that a non-weight-bearing low-intensity intradialytic exergame is feasible during routine hemodialysis treatment [112]. However, current exergames often fail to provide a sufficient extent of individualization and recommended dosage of exercise required for effective and safe training in patients with chronic diseases. Future studies need to design further exergames that can offer exercise, matching intensity and duration with established exercise guidelines.

#### 4.1.2. Nutrition Management

Patients with DFU often have a history of uncontrolled diabetes with an elevated level of HbA1c [113]. Since nutrition plays a crucial role in regulating HbA1c levels; therefore, patients with diabetes must be mindful of their diet. In 2015, the CDC Diabetes Prevention Recognition Program fixed standards to recognize both in-person and online diabetes prevention program (DPP) interventions. An organization or intervention must show an average of 5% weight loss for the participants with prediabetes after attending at least four sessions or more sessions to gain CDC recognition. As of 3 March 2021, there were 73 online DPP interventions with full recognition or preliminary recognition by CDC [114]. Most online DPP interventions used technologies that provided educational content, digital tracking of personal health information, social support, and feedback from automated or live health coaches [115].

Mobile phone text messaging is a simple and widely available method to deliver DPP. In 2011, Quinn et al. demonstrated improved diabetic care and a substantial reduction in glycated hemoglobin levels over one year in patients with type 2 diabetes using mobile- and web-based messaging systems [116]. The messaging system allowed patients to enter diabetes self-care data (e.g., blood glucose level, caloric intake, or medications). In response, patients received automated, real-time educational, behavioral, and motivational messages specific to the entered data. In a comparative study, Fischer et al. demonstrated patients with prediabetes who received six text messages per week were able to lose an additional 2 lb at 12 months compared to controls [117]. Furthermore, numerous studies suggest that text messaging can encourage weight loss [118,119]. Although these studies were not designed to manage diabetes, weight loss is crucial for diabetes prevention, thus provide an appealing intervention.

While text messaging is promising, smartphone and web-based technologies can provide even more extensive interventions to prevent diabetes. In a 24-week diabetic prevention program, Michaelides et al. evaluated the Noom Weight Loss Coach mobile app’s outcomes with human coaching among 121 individuals with prediabetes [120]. The Noom Weight Loss Coach is a publicly available app that tracks caloric intake, physical activity level and provides evidence-based weight-loss strategies while keeping the user motivated with instant feedback. At 24 weeks, the average weight loss was 6.58% in starters (*n* = 43) and 7.5% in completers (*n* = 36). Participants showed a high engagement level, with 84% of the sample completing nine lessons or more (CDC standard). Many other smartphone- or web-based applications (such as Fooducate or Lose It) can provide extensive nutrition information to educate patients to manage diabetes.

To enhance patient’s participation and engagement in online DPP interventions, Sepah et al. developed an online social-networking platform called Prevent that connected people with prediabetes. Prevent included four major intervention components: small-group support, health coaching, DPP curriculum, and digital tracking tools (Sepah et al., 2014). In this DPP, small groups of 10 to 15 participants were formed who were matched for demographically matched. A health coach served as a moderator to lead the DPP, and participants could interact with the health coach and other participants via private messages or telephone calls [121]. Among 187 participants who met CDC criteria (>4 sessions of Prevent) showed a 5% reduction in body weight with a 0.37% reduction in hemoglobin at 12 months [121]. In another study, Castro et al. showed 92% adherence in 501 patients with prediabetes for DPP based on Prevent while participants lost an average of 8% of their weight at 6 months and 7.5% of their weight at 12 months follow-up [122].

While smartphone and web-based programs can connect patients with nutritionists or dieticians, full automation may decrease the resources required to deliver a successful DPP. For instance, Ravana et al. developed a web-based diet system that takes advantage of AI for proposing diet plans to the public based on their health status and goals [123]. While various AI-based applications are coming up, research is needed to determine their efficacy compared to in-person DPP outcomes.

#### 4.1.3. Stress Management

Psychological distress (i.e., depression, anxiety, or stress) is highly prevalent in patients with diabetes than the general population [124] and is associated with adverse health outcomes [125]. However, it is important to understand that, unlike psychological distress, diabetes-specific distress, also referred to as diabetes burnout, occurs due to an emotional state where individuals experience guilt, self-denial, or the burden of complex self-management of diabetes itself [126]. Studies have reported that depressed patients with diabetes demonstrate poor self-management (i.e., adherence to diet plans, exercise regimens, and regular monitoring of blood glucose) [124,127], increasing the risk of DFUs. Furthermore, diabetes-related complications can elicit negative emotions such as stress which is often linked with impaired glycolytic metabolism and can delay wound healing. Therefore, it is important to identify technologies, which can mitigate psychological symptoms and assist in stress management among patients with diabetes.

During most of the in-person visits psychological well-being of the patients is determined using assessment tools (e.g., Patient Health Questionnaire-9 (PHQ-9) or Diabetes Distress Scale-17 (DDS-17)) in the standard paper format. However, this method is susceptible to error due to social desirability-, recall-, and self-reported biases [128]. Recently, mHealth based technologies have shown the potential to determine the patients’ psychological well-being through longitudinal evaluation [129,130]. For instance, three studies showed the feasibility of smartphones and the internet to send tailored text messages or structured emails to determine psychosocial outcomes in patients with diabetes [131,132,133]. Furthermore, various studies have demonstrated that automated telephonic assessment (ATA) can facilitate communication between health care providers and patients with diabetes, which can boost self-care, improve glycemic control, and mitigate diabetic-related distress [134,135,136,137]. ATA is a systematic collection of data and providing guidance based on patients’ conditions and preferences via telephone or the internet [126]. The main advantages of ATA over conventional face-to-face interaction involve simplicity, anonymity, low costs, and better self-reporting.

Moreover, studies have shown wearable technologies can provide digital biomarkers (such as heart rate, heart rate variability, blood pressure, galvanic skin response, and reduced physical activities), which can quantify stress and are less likely to be influenced by social and self-reported biases introduced from questionnaires [138,139].

### 4.2. Technologies to Promote Self-Risk Management

In patients with diabetes, self-risk management is referred to as taking personal responsibility for the health and organizing activities to minimize risk for DFUs. The Internet of medical things (IoMT) based technologies are empowering patients or their caregivers to boost self-risk management. The IoMT designates the interconnection of communication-enabled medical-grade devices and their integration to wider-scale health networks to improve patients’ health. Recent advancement in cloud data storage, automated monitoring, and connectivity to users and healthcare providers has facilitated in advancing IoMT which led the med-industry on the cusp of a home-care revolution.

Although IoMT based applications to manage DFUs are in an early stage, IoMT holds great potential to empower patients to take care of their health and promote patients’ central role. Voice-controlled IoT is already ubiquitous and is getting more ranging from intelligent personal assistance such as Apple’s Siri, Amazon Alexa, Google’s Google Now, and Microsoft’s Cortana. These devices adapt to the person’s voice and create an interface where that voice can reliably interact with various applications. For instance, with the help of automation, patients can be prompted to check their feet, glucose level or weight, and enter results into mobile patient portals. Moreover, patients or caregivers can transmit the results to their doctors in real-time. These fast-growing, low-cost, and widely available resources can help predict one’s risk for foot ulcers, infections, peripheral arterial disease, frailty, and other diabetes-associated complications, ultimately saving limbs and lives. While IoMT is being celebrated as the future of medicine, there are still some concerns that need to be addressed on patient compliance, battery life issues, and security and privacy. Nonetheless, we find ourselves in the early stages of a dramatic change in health care: where the merger of consumer electronics and medical devices has made the home the clinic of the future.

### 4.3. Technologies to Reinforce Adherence to Offloading

Empowering patients or their caregivers to take care of their health is a vital part of diabetes management. For instance, patients with an active DFU or a history of a DFU are often prescribed footwear or casts to reduce pressure on the affected area of the foot. However, wound management often fails due to poor adherence to the offloading provided by these devices. Recently, an offloading device, Motus Smart Boot, equipped with sensors and connected through the cloud, was introduced to promote patient adherence [56]. This device results from a collaboration between Optima Molliter (Civitanova, Italy), a leading orthopedic medical footwear manufacturer, and Sensoria Health (Redmond, Washington), a leader in embedded sensors and mobile applications for remote monitoring. The Motus Smart Boot facilitates the real-time assessment of offloading, managing weight-bearing physical activity dosage, and sending alert messages to patients, caregivers, and clinicians by applying AI algorithms to the sensor data. However, the clinical validity of this offloading is unclear at the time of drafting this review.

Another advancement related to the emergence of smart-flexible sensors implanted in insoles or socks combined with digital health apps has paved the way for monitoring PTS during daily living activities. In 2017, Najafi et al. examined adherence to alert-based cues for plantar pressure offloading in 17 patients with diabetic foot disease [140]. The real-time alerting system (the SurroSense Rx, Orpyx Medical Technologies Inc., Calgary, Canada) used a smartwatch for cueing based on pressure information from the plantar surface foot, which was wirelessly transmitted to the smartwatch. Participants were alerted for offloading by providing simple instructions via smartwatch (e.g., walk few steps after prolonged sitting or standing, etc.) to manage unprotected sustained plantar pressures in an effort to prevent foot ulceration. Participants used the SurroSene system for three months, and successful response to an alert was defined as pressure offloading of the area with sustained pressure within 20 min of detection. Patient adherence, defined as daily hours of device wear, was determined using sensor data and patient questionnaires. They observed that participants receiving at least one alert every two hours wore their offloading for longer and had better adherence in responding to alerts.

Furthermore, various websites and smartphone apps are available which can provide nutritional education, motivational support for behavioral reinforcement, and track blood sugar. Many patients access the Internet to gather facts to understand better what diabetes is, how it is treated, and what they can or should eat. Providing patients with a few key high-quality websites can facilitate “content management,” as will discussing the content they obtained so clarifications can be provided as needed. Therefore, adherence to a healthy lifestyle (i.e., higher physical activities, healthy diet, or regular medication) and regular blood glucose monitoring can prevent DFUs.

Many smartphone applications increase mindfulness in making daily life decisions, the discipline for self-management and self-care, and healthy habits. In a 13 week-long study, Dugas et al. used a gamified-smartphone app (DiaSocial) to promote adherence to physical exercise, improved nutrition, and adherence to drug therapy among 27 veterans (mean age 65.4 years) with DM2 [141]. The app gave reward points based on the participant’s healthy behavior manifestation (i.e., maintaining blood glucose level, recording blood glucose level, healthy nutritional intake, minutes of exercise, and medication adherence). Results showed that stronger adherence was associated with higher locomotion and greater reduction of HbA1c. The mean HbA1c in the CG, which received a traditional health education of recommendations and control, was 8.78, and in the IG, it was 8.06.

## 5. Conclusions

Long-term medical management to reduce the risk of ulcer and/or its recurrence is important to reduce the global diabetic foot disease burden. The increasing development and use of technology within every aspect of our lives represent an opportunity for creative solutions to prevent or better manage diabetic foot problems. In particular, recent advances in wearable and mobile health technologies appear to show promise in measuring and modulating dangerous foot pressure and inflammation to extend remission and improve the quality of life for these most complex patients. Thanks to the new “smart” sensors and communications technology available today, new opportunities have opened to smartly manage DFUs with personalized screening and timely intervention. With the help of automation and AI, innovative solutions are emerging to facilitate the provision of comprehensive and easy to execute feedback (e.g., gamification to engage patients and improves adherence to offloading), recommendation (e.g., personalized and easy-to-understand guidelines to manage DFUs), and notification (e.g., alerting about signs of inflammation, which precede skin breakdown). These technologies can be used as supportive care to empower patients in self-care. For example, everyday measurement and timely feedback of inflammatory response (e.g., change in plantar temperature as a sign of inflammation), high plantar stress (e.g., change in plantar pressure as signs of the formation of callus, presence of a foreign object inside of shoes, infective offloading, or altered lower extremities biomechanics because of aging and/or diabetes), and daily activities (e.g., prolonged unbroken walking and standing) have been shown to be effective in improving the management of DFUs. They can transmit the results to their doctors in real-time. These fast-growing, low-cost, and widely available resources, can help predict one’s risk for foot ulcers, infections, peripheral arterial disease, frailty, and other diabetes-associated complications, ultimately saving limbs and lives.

More importantly, with advances in technologies, the model of care delivery could be shifted from considering a patient as a passive recipient of care to an active engagement in the health ecosystem. The COVID-19 pandemic, in particular, accelerated the adoption of such a new model of care delivery to facilitate remote care delivery to patients and empower them in self-care. For instance, because people with diabetes represent a fragile population, it is recommended to avoid unnecessary diabetes-related hospital admissions to reduce the risk of COVID-19 exposure in the hospital. This is disrupting the best practices for preventing DFU. It is estimated that DFU patients are seen in outpatient facilities 14 times per year and hospitalized approximately 1.5 times per year. As health care providers searching for alternatives to deliver timely care to patients with the presence or risk of DFU, it is tempting to imagine a post-COVID future may lead to some positive changes in healthcare for people with chronic illness, particularly in promoting preventive and personalized care for people with DFU.

## Figures and Tables

**Figure 1 medicina-57-00377-f001:**
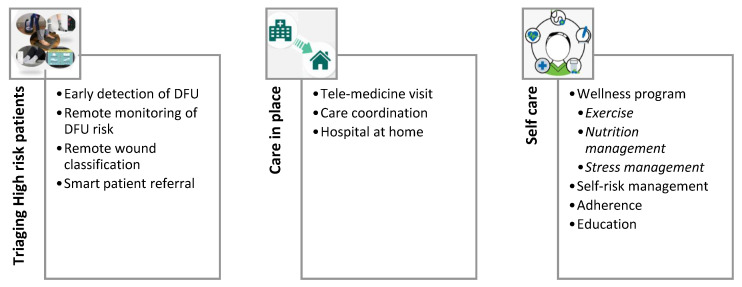
Recent developments in three specific areas are discussed (shown in detail in Figure 1): (1) Triaging high-risk patients for timely intervention, (2) Care in place to avoid frequent hospital visits, and (3) Self-care to empower patients and their caregivers. DFU: diabetic foot ulcer.

**Figure 2 medicina-57-00377-f002:**
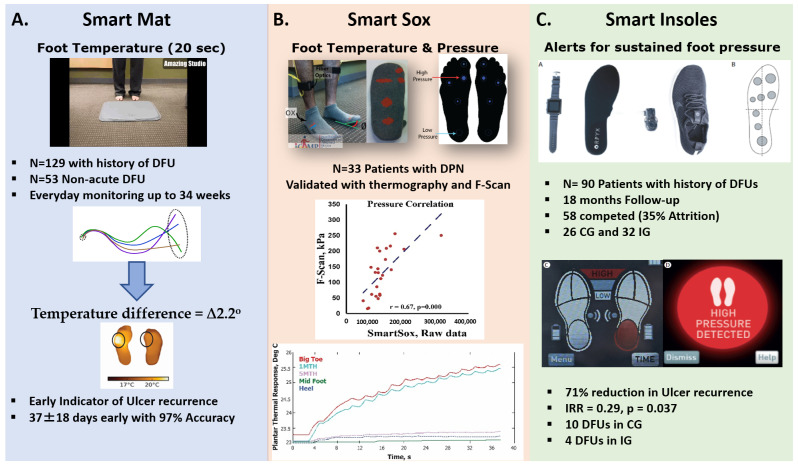
Recently, several digital health technologies are developed to screen digital biomarkers of diabetic foot ulcer (DFU). Three typical examples of these technologies are illustrated in this figure: (**A**) Frykberg et al. (2017) suggested that a smart mat (Podimetrics Mat) enables remote monitoring of plantar temperature and provides a digital biomarker of inflammation which may predict prospective DFU with 37 days lead time and 97% accuracy [16]. The specificity was, however, low (57%), indicating a high false-positive rate. Despite poor specificity, 37 days lead time and high accuracy could make this remote patient monitoring platform a practical device for triaging high-risk patients and timely preventive care. (**B**) Najafi et al. (2017) proposed the concept of SmartSox, which enables continuous screening of the digital biomarkers of DFU, including plantar pressure, temperature, and big toe range of motion [27]. Their results show an acceptable agreement between measured digital biomarkers and gold standards. However, the study is limited to a cross-sectional study, and thus the validity of this technology to predict DFU is unclear. (**C**) Abbott et al. (2019) tested the efficacy of a combination of smart insole and smartwatch that continuously monitor plantar pressure and alert patient via smartwatch when a sustained pressure (pressure lasting longer than 15 min) was detected [28]. In proof of concept randomized control trials, they suggested that such technique may reduce the recurrence of ulcers up to 71%. However, due to the small sample size and high dropout rate, the conclusion of this study is limited. Abbrevietions: (CG) control group, (IG) Intervention group, (DPN) diabetes related pheripheral neuropathy, and (IRR) Incidence rate ratio.

**Figure 3 medicina-57-00377-f003:**
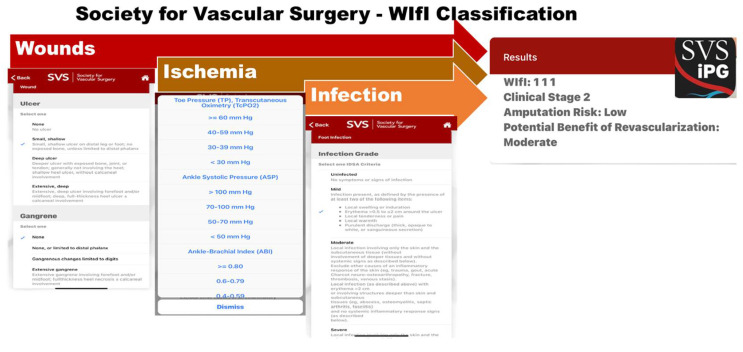
The Society for Vascular Surgery WIfI classification application (SVG iPG). The SVS iPG application provides education on different types of wounds and classifies wound based on three major factors that impact amputation risk. It also includes a comprehensive recommendation to manage each wound type including the need to make a referral for revascularization. The WIfi classification is based on simple and easy to collect info including wounds characteristics (size and presence of gangrene), ischemia (assessed using one of the three vascular metrics: Toe Pressure, Ankle Systolic Pressure, or Ankle Brachial Index), and degree of infection.

**Figure 4 medicina-57-00377-f004:**
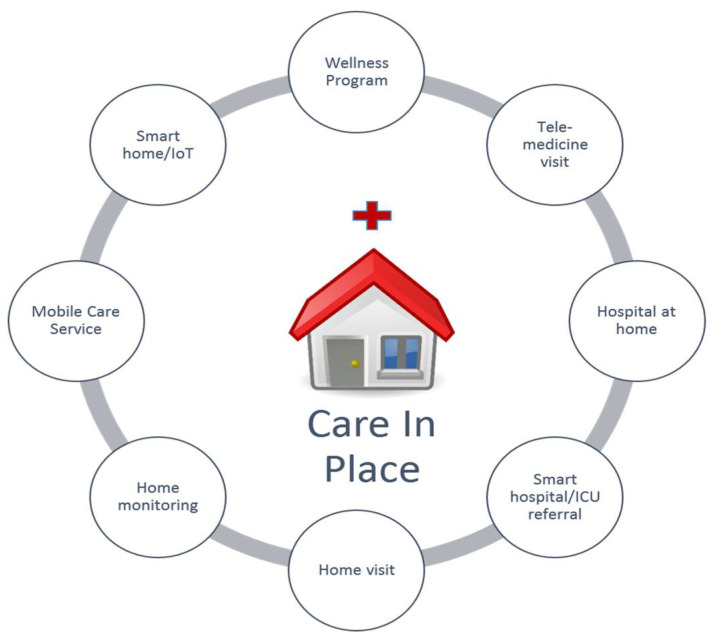
Care in place: Healthcare delivery has traditionally been designed around the provider, located in hospitals, clinics, or doctors’ offices. With advances in telecommunication and telehealth technologies, new solutions have emerged facilitating “care in place” toward settings where the care goes to the patient instead of the patient going to the care.

**Figure 5 medicina-57-00377-f005:**
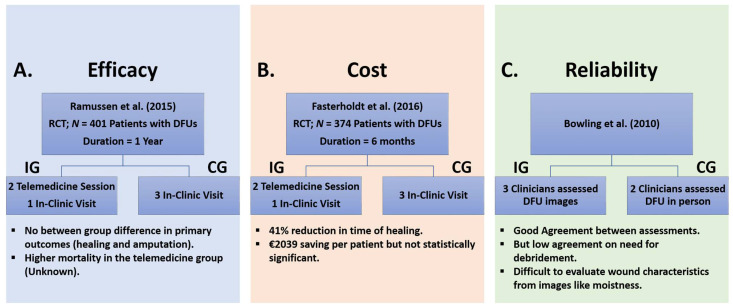
Recent studies support the efficacy, cost-effectiveness, and reliability of telemedicine support for DFU management. (**A**) In 2015 Rasmussen et al. demonstrated that by replacing 2/3 of in-clinic visits with telemedicine visits, the main outcomes (healing and amputation) would be similar to having all clinic visits [57]. (**B**) In 2018, Fasterholdt et al. showed that replacing face-to-face visits with telemedicine visits leading to a 41% reduction in total staff time used on outpatient consultations, which lead to noticeable cost-saving (approximately EUR 2039 for a patient) [58]. However, this reduction was not statistically significant in their sample (**C**) In 2011, Bowling et al. demonstrated a good agreement between assessment of 3D wound pictures and face-to-face assessment of wounds except for cases that needed debridement [59].

## Data Availability

Not applicable.

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
