# Peer review of "Harnessing Digital Health Technologies to Remotely Manage Diabetic Foot Syndrome: A Narrative Review"

_medicina, 2021, doi:10.3390/medicina57040377_

Round 1
Reviewer 1 Report
This is a review paper where authors focusing on the technologies that can facilitate care in place or empower self-care for patients with an active or high risk of diabetic foot ulcers (DFU). The paper is well written, the text is clear and easy to read. Authors structured the review paper in three important topics to take into consideration for clinicians that are treating patients with DFU, specially in COVID pandemic and for future.
1. Triaging high-risk patients for timely intervention
Early detection of DFU
Remote monitoring of DFU risk
Remote monitoring of wound characteristics
Smart patient referral
2. Care in place to avoid frequent hospital visits
Telemedicine visits
Care Coordination
Hospital at home
3. Self-care to empower patients and their caregivers.
Wellness program: Exercise, Nutrition management, Stress management
Technologies to promote self-risk management
Technologies to reinforce adherence to offloading
The evidence (149 references) and arguments used in the develope of this review are consistent and conclusion is according with the evidence presented.
In my personal point of view this review paper is very useful for open eyes for many clinicians to know a new model of care delivery to facilitate remote care delivery and empower patients in self-care, specially in people with diabetes that represent a fragile population.
Author Response
We deeply appreciate the positive and encouraging notes. We are very happy to hear that from point of view of the reviewer "This review paper is very useful for open eyes for many clinicians to know a new model of care delivery to facilitate remote care delivery and empower patients in self-care, specially in people with diabetes that represent a fragile population.". This is exactly our objective and we are hoping this review paper could enhance awareness in the field and encourage more research or adoption of new technology to address diabetic foot and its devastated condition.
Reviewer 2 Report
The topic of this study is trend, so it may be helpful to the readers.
Title is right, but I recommend change “Diabetes Foot Syndrome” to Diabetic Foot Syndrome.
Abstract is well, but I recommend not to use abbreviations in abstract section.
Introduction section is deep enough with and adequate focus that may help readers to improve knowledge about the topic. Please, in line 52 use DFU as abbreviation instead of use diabetic foot ulcer.
In general, this is a well-written manuscript and it is well structured with different sections with an enough number of figures that help to achieve a better understanding of the review. However, I have found multiple spelling errors that should be corrected:
-Line 182: lessor toes, please change to lesser toes.
-Line 192: ulcer reocurrence, please change to ulcer recurrence.
-Line 254: comhrehensive, please change to comprehensive.
-Line 256: charcetreistics, please change to characteristics.
-Line 257: gangarne, please change to gangrene.
-Line 294: diabetic feet, please change to diabetic foot.
-Line 306: havemerged, please change to have emerged.
-Line 345: €2039, please change to 2039€.
-Line 494: please erase ).
-Line 543: HbAc1d, please change to HbA1c.
-Line 715: reocurrence, please change to recurrence.
Due to this multiple errors found in the manuscript, I recommend an English re-editing.
Conclusions are supported by the review data, but I recommend not to use references in conclusions section, because in a narrative review the conclusions should be perform by the authors of the review and should not use literally the conclusions of other authors previously publish.
Author Response
We appreciate the encouraging and constructive comments made by the reviewer. In particular, we deeply appreciate the constructive critiques and apologize for typos and grammatical mistakes. We have carefully proofread the manuscript to accommodate the reviewer's suggestions and ensured to avoid any other typo or spelling mistakes. In the attached we addressed point to point comment and would be happy to address any additional concerns.

Round 2
Reviewer 2 Report
The authors have answered all my comments and correct all the errors in the manuscript.
Thank you.